# The Influence of Acute Sprint Interval Training on Cognitive Performance of Healthy Younger Adults

**DOI:** 10.3390/ijerph19010613

**Published:** 2022-01-05

**Authors:** Fabian Herold, Tom Behrendt, Caroline Meißner, Notger G. Müller, Lutz Schega

**Affiliations:** 1Research Group Degenerative and Chronic Diseases, Movement, Faculty of Health Sciences Brandenburg, University of Potsdam, Karl-Liebknecht-Str. 24–25, 14476 Potsdam, Germany; fabian.herold@fgw-brandenburg.de (F.H.); notger.mueller@uni-potsdam.de (N.G.M.); 2Department of Neurology, Medical Faculty, Otto von Guericke University Magdeburg, Leipziger Str. 44, 39120 Magdeburg, Germany; 3Research Group Neuroprotection, German Center for Neurodegenerative Diseases (DZNE), Leipziger Str. 44, 39120 Magdeburg, Germany; 4Department of Sport Science, Otto von Guericke University Magdeburg, Zschokkestr. 32, 39104 Magdeburg, Germany; caroline.meissner@st.ovgu.de (C.M.); lutz.schega@ovgu.de (L.S.); 5Center for Behavioral Brain Sciences (CBBS), Brenneckestraße 6, 39118 Magdeburg, Germany

**Keywords:** sprint interval training, acute exercise, cognition, lactate, exercise–cognition

## Abstract

There is considerable evidence showing that an acute bout of physical exercises can improve cognitive performance, but the optimal exercise characteristics (e.g., exercise type and exercise intensity) remain elusive. In this regard, there is a gap in the literature to which extent sprint interval training (SIT) can enhance cognitive performance. Thus, this study aimed to investigate the effect of a time-efficient SIT, termed as “shortened-sprint reduced-exertion high-intensity interval training” (SSREHIT), on cognitive performance. Nineteen healthy adults aged 20–28 years were enrolled and assessed for attentional performance (via the d2 test), working memory performance (via Digit Span Forward/Backward), and peripheral blood lactate concentration immediately before and 10 min after an SSREHIT and a cognitive engagement control condition (i.e., reading). We observed that SSREHIT can enhance specific aspects of attentional performance, as it improved the percent error rate (F%) in the d-2 test (*t* (18) = −2.249, *p* = 0.037, d = −0.516), which constitutes a qualitative measure of precision and thoroughness. However, SSREHIT did not change other measures of attentional or working memory performance. In addition, we observed that the exercise-induced increase in the peripheral blood lactate levels correlated with changes in attentional performance, i.e., the total number of responses (GZ) (r_m_ = 0.70, *p* < 0.001), objective measures of concentration (SKL) (r_m_ = 0.73, *p* < 0.001), and F% (r_m_ = −0.54, *p* = 0.015). The present study provides initial evidence that a single bout of SSREHIT can improve specific aspects of attentional performance and conforming evidence for a positive link between cognitive improvements and changes in peripheral blood lactate levels.

## 1. Introduction

There is growing evidence in the literature that a single bout of physical exercises can acutely enhance performance in cognitive domains such as attention or executive functioning [1,2]. However, the optimal exercise characteristics (e.g., type of physical exercise, exercise intensity, and exercise duration) to effectively improve cognitive performance are largely unknown [1,2]. Among various types of physical exercise, many studies have shown that moderate-intensity exercises can acutely increase cognitive functions [1,3]. However, moderate-intensity exercises often require a higher time commitment, which is, among other factors, known to be one of the main perceived barriers preventing younger individuals (e.g., younger sedentary people) [4,5,6] and the general population [7,8,9] from performing physical exercises, even though there is the possibility that “lack of time” can be both an excuse or a barrier hindering an individual from being physically active [6].

In recent years, sprint interval training (SIT) has been promoted as a time-efficient alternative or additional approach to moderate-intensity exercises to improve health [10]. SIT is a special form of high-intensity training (HIT), which is typically characterized by short working bouts of “all-out” efforts (≤30 s) that alternate with longer recovery bouts (2–4 min) at relatively low intensity [11,12,13]. The commonly used protocol for SIT is the so-called Wingate protocol, which consists of four bouts of 30-s “all-out” effort sprints intermitted by 4 min of recovery [10]. In this context, it has been noticed that, in younger adults, such an SIT protocol (i.e., Wingate protocol, three days per week) can be a promising alternative to a classical endurance training (i.e., continuous cycling at a workload that elicits ∼65% of peak oxygen consumption (VO_2peak_) for 40–60 min, five days per week), given the evidence that both training protocols (being conducted for eight weeks) lead to comparable changes in the parameters of muscle metabolism, but SIT needs a remarkably lower time commitment (∼90 min vs. ∼270 min per week) [14]. In a comparable manner, it was noticed that, in sedentary younger adults, three weekly sessions of SIT (i.e., 3 × 20 s sprints, 120 s recovery, ~30 min per week) or moderate-intensity endurance training (i.e., 45 min of continuous cycling at ~70% maximal heart rate, 150 min per week) lead to comparable changes in the cardiometabolic parameters (e.g., VO_2peak_ and muscle mitochondrial content) [15]. With respect to cognitive performance, the influence of SIT has not been extensively studied, but some acute exercise–cognition studies suggest that in a cohort of adolescents [16] and younger adults [17] SIT can increase the performance in tasks probing attention or executive functioning.

However, although the above-presented evidence indicates that SIT using the Wingate protocol is a genuinely time-efficient and effective exercise modality, the Wingate protocol is considered as physically and mentally very demanding and, thus, is not well-situated to be applied in psycho-physiologically impaired cohorts (e.g., sedentary individuals and older adults) [10,18]. Hence, it has been proposed that future studies should focus on the investigation of protocols with fewer sprint repetitions and/or lesser sprint time duration [10,18]. In line with this purpose, the findings of a recent study suggest that, in younger adults, a reduction of the sprint time from 20 s to 5 s is favorable when psychological parameters such as affect, effort, and enjoyment were considered [19]. In a comparable manner, another study reported that younger adults rated SIT exercise sessions with a 5:40 protocol (24 × 5 s “all-out” effort sprints, 40 s recovery) as more enjoyable than an SIT exercise sessions using a classical 30:240 Wingate protocol (4 × 30 s “all-out” effort sprints, 240 s recovery) [20]. Moreover, the younger adults were more confident about completing and expressed greater intention to engage in the 5:40 protocol as compared to 30:240 protocol [20]. Collectively, the above-mentioned evidence suggest that shortened SIT protocols (e.g., 8 × 5 s), being termed as “shortened-sprint reduced exertion high-intensity training” (SSREHIT), are, from a psychological point of view, more suitable for an exercise intervention than the classical SIT protocol (i.e., Wingate protocol). Given the time-efficiency of SIT (as compared to continuous moderate-intensity exercises) in general and the better psychological characteristics of SSREHIT (as compared to classical SIT using the Wingate protocol) in particular, SSREHIT can be considered as a promising exercise modality being worth future investigations [19,21,22]. Interestingly, the current research has demonstrated that SSREHIT elicits substantial benefits in markers of cardiorespiratory fitness and metabolic health in healthy people [22,23,24,25,26,27,28]. However, to the best of our knowledge, there is, so far, no study available that has investigated the acute effect of SSREHIT on cognitive performance (e.g., attentional performance). From an applied perspective, a deeper understanding to which extent SIT might influence cognitive performance can be of great interest for different settings with relatively high cognitive demands and a lack of time for physical exercise breaks (e.g., office workers, assembly line workers, and university students). Based on accumulating evidence showing that acute physical exercises can influence performance in a broad range of cognitive domains positively [1,2], we hypothesized that SSREHIT can improve cognitive performance (e.g., attentional performance) as well.

In this context, this study also seeks to investigate the neurobiological processes that drive possible exercise-related changes in cognitive performance in response to acute physical exercises (i.e., SSREHIT) because the knowledge in this direction is relatively limited [2]. In the literature, there are several hypotheses on possible neurobiological processes (e.g., the catecholamine hypothesis [29,30] or the interoception model [31]) that drive the changes in cognitive performance in response to an acute bout of physical exercises (for review, see Reference [2]). Among those neurobiological processes, there is some evidence that exercise-related changes in the blood lactate levels can be an important factor [32], given the findings of some studies reporting in younger adults a positive association between changes in the levels of blood lactate and cognitive performance (i.e., executive functioning) in response to an acute bout of physical exercises [32,33]. To further investigate the link between lactate and cognitive performance, in this study we will assess the relationship between exercise-related changes in the levels of peripheral blood lactate concentration and measures of cognitive performance (i.e., attentional performance and working memory performance). Based on available studies [32,33], we hypothesize that a positive association between exercise-related changes in the levels of blood lactate concentration and cognitive performance exists.

In summary, the purpose of the current study is twofold, as: (i) we aim to investigate to which extent an acute bout of SSREHIT (i.e., 6 × 6-s sprints, 60-s recovery between sprints [26,27,28,34]) changes measures of cognitive performance (i.e., attentional performance and working memory performance) in a cohort of healthy younger adults, and (ii) we aim to elucidate whether exercise-related changes in the blood lactate levels are linked to changes in our measures of cognitive performance.

## 2. Materials and Methods

### 2.1. Participants

We recruited 19 healthy young adults (11 female/8 male; age: 22.7 ± 2.3 years; body height: 176.0 ± 12.7 cm; body mass: 68.1 ± 11.0 kg). Please note that, initially, 20 participants were recruited, but one participant dropped out due to personal reasons.

All of the enrolled participants met the following inclusion criteria: (i) age between 18 and 30 years; (ii) right-handed (assessed via Edinburgh Handedness Inventory (EHI)—a score of ≥50 indicates right-handedness [35]); (iii) absence of a depression (assessed via Becks Depression Inventory (BDI-II)—cut-off score to be included ≤ 13 [36]); (iv) absence of musculoskeletal, cardiovascular, endocrinological, psychiatric, and/or neurological disorders (assessed via self-reports); and (v) low-risk status for physical exercise-related adverse events (assessed via Physical Activity Readiness Questionnaire (PARQ) [37]). All individuals who did not meet our inclusion criteria were excluded from this study.

The study was approved by the local ethics committee of the Medical Faculty of the Otto von Guericke University Magdeburg (97/20) and was prospectively registered in the German Clinical Trial Register (DRKS00022577; 4 August 2020). All study procedures were conducted in accordance with the latest version of the Declaration of Helsinki, and all participants provided written informed consent to participate.

### 2.2. Study Design

The participants were asked to visit our laboratory three times at the same time of the day (±60 min). The visits were separated by at least five days.

At the first visit, the participants were informed about the study procedures, were screened for eligibility (see Section 2.1 Participants), and the eligible participants were asked to complete the following questionnaires: (i) level of education (via self-reports), (ii) a physical activity questionnaire (BSA; derived from the German Bewegungs-und Sportaktivitatsfragebogen) [38], and (iii) the Pittsburgh Sleep Quality Index (PSQI) [39]. Given the evidence (i) that, in younger adults, both the regular level of physical activity and measures of sleep are associated with cognitive performance (i.e., inhibitory control) and (ii) that sleep efficiency mediates the relationship between regular physical activity and cognitive performance (i.e., inhibitory control) [40], the BSA and the PSQI were assessed to provide a more comprehensive overview on the characteristics of the included participants (see Table 1). In addition, the participants performed two familiarization 6-s “all-out” effort sprint trials during the first visit. All physical exercise conditions were performed on a magnetically and air-braked cycle ergometer (Wattbike Pro, Wattbike, UK), which supports a power range from approximately 50 to 3760 Watts (W).

At the second and third visits, the participants completed the cognitive tests immediately before and 10 min after an acute bout of SSREHIT (SSREHIT condition) or 20 min of seated rest (control condition). A passive rest period of 10 min after the SSREHIT condition was used because a meta-analysis reported that the largest effects of vigorous physical exercises (e.g., “all-out” efforts, as in sprint interval training) on cognitive performance can be observed when such a time delay after exercise cessation is considered [1]. As recommended in the literature [2,42], this study was conducted in a within-subject crossover design with both pretest and posttest assessments (see Figure 1), and we randomized the order of the SSREHIT and the control condition using a specific software program (RITA version 1.51, Evidat, Germany). Each participant performed each condition (i.e., REHIT condition or control condition) once.

### 2.3. Study Conditions

The SSREHIT condition started with a standardized 3-min-long warm-up period on the Wattbike set at the resistance level of 1 for both the magnetic- and air-braked resistance at ~60–80 revolutions per minute (rpm) with two acceleration phases of ~3 s at 60 and 120 s. Afterward, all participants performed six 6-s “all-out” effort sprints on the Wattbike ergometer, which were intermitted with passive rest periods of 1 min [26,27,28,34]. Verbal encouragement was given during all sprints. Based on previous studies [43,44,45,46,47], the resistance of the Wattbike was set to 1 and 10 for the magnetic- and air-braked resistances, respectively (equating to 704 ± 5.4 Watt at 110 revolutions per minute [48]). As frequently done in comparable studies (for review, see Reference [2]), in the control condition, the participants were asked to sit quietly in a comfortable chair and read for the same period of time (20 min, also known as cognitive engagement control [2]).

In addition, all participants were instructed (i) to avoid strenuous physical activities 48 h before each experimental session, (ii) to abstain from excessive alcohol or caffeine consumption for 24 h before each experiment, (iii) to keep their normal sleep rhythm, and (iv) to consume their last meal at least 3 h before the start of the experimental conditions.

### 2.4. Outcomes

#### 2.4.1. Primary Outcomes—Measures of Cognitive Performance

In this study, the paper–pencil version of the d2 test measuring selective attention and concentration performance was used [49]. The d2 test consists of 14 lines, and each line entails a string of 47 randomly mixed letters (i.e., “d” and “p”). Each letter is flanked by dashes (i.e., individually or in pairs above and/or below the letters). The participants are instructed to mark within 20 s all “d’s” in a line that are flanked by two dashes, which may be arranged individually above and below or in pairs above or below the “d”. After 20 s, the participant is advised by the experimenter to continue with the next line of letters (the whole test lasts 280 s). The performance in the d2 test was assessed by using: (i) the total number of responses (German: “Gesamtzahl aller bearbeiteten Zeichen”; GZ), including the correct responses and mistakes in the d2 test—a quantitative measure of working speed, (ii) the standardized number of correct responses minus errors of commission (German: “Standardwert der Konzentrationsleistung”; SKL)—an objective measure of concentration, and (iii) the number of all errors (error of omission + error of commission) related to the total number of responses (German: “Fehlerprozentwert”; F%)—a qualitative measure of precision and thoroughness. Errors are defined as errors of omission (number of correct responses (i.e.,“d” with two dashes) missed) and errors of commission (any distractor items such as a “p” or a “d” with one dash or more than two dashes incorrectly marked) [50,51,52]. The d2 test has an excellent reliability [49] and has been frequently used to quantify exercise-induced changes in attentional performance [50,51,53].

Digit Span Forward (DSF) and Digit Span Backward (DSB) were used to assess working memory performance, as done in previous acute exercise–cognition studies [54,55,56]. In this cognitive test, the participant has to listen to sets of ascending digit numbers read out loud at a pace of one digit per second. After listening, the participant was asked to recall the sequence of given numbers out loud (in DSF, in the same order and, in DSB, in the reverse order). Both DSF and DSB start with a sequence of two numbers (e.g., 7–3). For each span, two trials are performed, and the span is gradually increased in the step of one item. Each time, a different sequence of numbers is used. The test is stopped after two consecutive fails on the same set of items [55,56]. Each correct answer is scored with one point, so the maximal scores are 28 and 20 for DSF and DSB, respectively. For all cognitive tests, parallel versions were used to reduce the influence of learning (recall) effects.

#### 2.4.2. Secondary Outcomes—Psychological Parameters

As shown in Figure 1, the following psychological parameters were quantified: (i) affective responses via the 11-point Feeling scale (FS), which ranges from −5 (i.e., “very bad”) to +5 (i.e., “very good”) [57,58]; (ii) relative perceived exertion (RPE) via a 15-point Borg scale, which ranges from 6 (i.e., no exertion) to 20 (i.e., maximal exertion) [59]; and (iii) changes in concentration, motivation, and mental fatigue via visual analog scales, which range from 0 mm (i.e., not at all) to 100 mm (i.e., extremely) [33,60].

#### 2.4.3. Secondary Outcomes—Physiological Parameters

The peripheral blood lactate concentration was measured using blood samples from the ear lobe of the participants (immediately before cognitive testing and after exercise cessation; see Figure 1 for time points of the assessments). The time points of the lactate assessments are oriented on previous and comparable studies in the field measuring the peripheral blood lactate concentration before cognitive testing and after exercise cessation [33,60]. All blood samples were analyzed using the Super GL3 device (HITADO, Möhnesee, Germany; measuring error < 1.5%, according to the manufacturer’s information).

### 2.5. Statistical Analysis

The statistical analysis was performed with JASP (version 0.14.1.0, JASP Team, Amsterdam, The Netherlands). The delta scores (values of posttest minus values of pretest, separately calculated for each condition and each participant) of the cognitive tests (d2, DSF, and DSB) were normally distributed (verified by the Shapiro–Wilk test and/or visual inspection of the data), and thus, paired *t*-tests and Cohen’s d (with 95% confidence intervals (CI)) were calculated to compare the delta scores of the exercise and control condition. We rated Cohen’s d as follows: small effect <0.2, medium effect ≥0.2 to ≤0.8, and large effect >0.8 [61,62].

The data of the RPE scale, FS scale, VAS, and peripheral blood lactate concentration were not normally distributed, and thus, the nonparametric Friedman test and Conover post hoc tests were used to analyze the possible time effect. The adjustment of the alpha level of the post hoc test was conducted using Holm correction (p_holm_) [63]. To compare the difference between the exercise and control conditions, Wilcoxon tests were conducted, and the effect size r (with 95% CI) was computed. We rated the effect size r as follows: <0.1: very small effect, ≥0.1 to ≤0.29: small effect, ≥0.3 to ≤0.49: medium effect, and ≥0.5: large effect [64,65,66,67,68].

In addition, as previous studies reported a relationship between changes in the peripheral blood lactate and cognitive performance [33,60], we conducted a repeated-measures correlation analysis between cognitive performance measures and peripheral blood lactate concentrations using a freely available R package [69]. The repeated-measures correlation coefficient (r_m_) was rated as follows: no correlation <0.19, low correlation ≥0.20 to ≤0.39, moderate correlation ≥0.40 to ≤0.59, moderately high correlation ≥0.60 to ≤0.79, and high correlation ≥0.8 [61,62]. The level of significance of all statistical tests was set to α < 0.05.

## 3. Results

### 3.1. Cognitive Performance

The participants reduced the number of all the errors (F% (*t* (18) = −2.247, *p* = 0.037, d = −0.516 (CI 95% −0.989 to −0.030)) to a greater extent in the SSRHEIT condition as compared to the control condition. Regarding GZ (*t* (18) = 1.495, *p* = 0.152, d = 0.343 (CI 95% −0.125 to 0.802)), SKL (*t* (18) = 1.899, *p* = 0.074, d = 0.436 (CI 95% −0.041 to 0.901)), DSF (*t* (18) = −1.707, *p* = 0.105, d = −0.392 (CI 95% −0.854 to 0.081)), and DSB (*t* (18) = −0.218, *p* = 0.830, d = −0.050 (CI 95% −0.499 to 0.401)), no statistically significant differences between the SSREHIT and the control conditions were observed. A descriptive overview on the cognitive performance measures is shown in Table 2.

### 3.2. Psychological and Physiological Parameters

A significant effect of time concerning the RPE ratings in the SSREHIT condition (Χ^2^ = 35.041 (*n* = 19; df = 2), *p* < 0.001) but not in the control condition (Χ^2^ = 2.000 (*n* = 19; df = 2), *p* = 0.368) was noticed. The post hoc tests indicated that, in the SSREHIT condition, the RPE rating was higher after exercise (T (36) = 5.883, p_holm_ < 0.001) and at the posttest (T (36) = 2.320, p_holm_ = 0.026) as compared to the pretest. Furthermore, the RPE rating in the SSREHIT condition was lower at the posttest as compared to after exercise (T (36) = 3.563, p_holm_ = 0.002).

The comparisons between both conditions revealed that the RPE ratings in the SSREHIT condition were higher after exercise (W (*n* = 19) = 190.000, *p* < 0.001, r = 1.000 (CI 95% 1.000–1.000)) and at posttest (W (*n* = 19) = 120.000, *p* < 0.001, r = 1.000 (CI 95% 1.000–1.000)) but not at pretest (W (*n* = 19) = 4.000, *p* = 0.789, r = 0.333 (CI 95% −0.704 to 0.917)).

Regarding the FS ratings, whether in the SSREHIT condition or in the control condition, a statistically significant time effect was observed (*p* > 0.05). However, at the following time points, the ratings in the FS were lower in the SSREHIT condition compared to the control condition: after exercise (W (*n* = 19) = 28.500, *p* = 0.013, r = −0.667 (CI 95% −0.870 to −0.271)) and at posttest (W (*n* = 19) = 4.000, *p* = 0.004, r = −0.912 (CI 95% −0.974 to −0.728)). No difference in the FS ratings were observed at pretest (W (*n* = 19) = 19.500, *p* = 0.070, r = −0.571 (CI 95% −0.853 to −0.033)).

There were no significant differences between the pretest and the posttest values of motivation, concentration, and mental fatigue (assessed via VAS) in the SSREHIT condition and control condition (*p* > 0.05). Furthermore, there was no difference in motivation, concentration, and mental fatigue between the SSREHIT condition and the control condition (*p* > 0.05).

Regarding the peripheral blood lactate concentration, a significant time effect in the SSREHIT condition (Χ^2^ = 38.000 (*n* = 19; df = 2), *p* < 0.001) and in the control condition (Χ^2^ = 7.891 (*n* = 19; df = 2), *p* = 0.019) was noticed. The post hoc tests indicated that, in the SSREHIT condition, the peripheral lactate concentration was higher after exercise (T (36) = 6.164, p_holm_
*p* < 0.001) and at posttest (T (36) = 3.082, p_holm_
*p* = 0.008) as compared to the pretest. Furthermore, the peripheral lactate concentration in the SSREHIT condition was lower at posttest compared to after exercise (T (36) = 3.082, p_holm_
*p* = 0.008).

In the control condition, the peripheral lactate concentration values measured at the posttest were lower than the values measured at the pretest (T (38) = 2.795, p_holm_ = 0.025).

In comparison to the control condition, the peripheral lactate concentration in the SSREHIT condition was higher after exercise (W (*n* = 19) = 190.000, *p* < 0.001, r = 1.000 (CI 95% 1.000–1.000)) and at posttest (W (*n* = 19) = 190.000, *p* < 0.001, r = 1.000 (CI 95% 1.000–1.000)) but not at pretest (W (*n* = 19) = 53.000, *p* = 0.453, r = −0.221 (CI 95% −0.654 to 0.321)). A detailed overview about the psychological and physiological measures is provided in Table 3.

In addition, in the SSREHIT condition, the following correlations were observed between changes in the peripheral blood lactate levels and measures of attentional performance: GZ (r_m_ = 0.70 (CI 95% 0.342 to 0.878), *p* < 0.001), SKL (r_m_ = 0.73 (CI 95% 0.391 to 0.891), *p* < 0.001), and F% (r_m_ = −0.54 (CI 95% −0.802 to 0.093), *p* = 0.015). In the control condition, no significant correlations between changes in the peripheral blood lactate levels and attentional performance were noticed (*p* > 0.05).

## 4. Discussion

In this study, we tested whether SSREHIT as a time-efficient exercise modality can improve cognitive performance (i.e., attentional performance and working memory performance) in younger adults. We observed that SSREHIT compared to the control condition improved F% (a qualitative measure of precision and thoroughness) but did not lead to an increase in GZ (a quantitative measure of working speed), SKL (an objective measure of concentration), or the number of correctly remembered items in DSF and DSB (quantitative measures of working memory performance).

The observed increase in attentional performance is consistent with the evidence provided in the literature [1,2,70], whereas the unaltered working memory performance does not fully match with the findings of two previous meta-analyses that reported exercise-related improvements in this cognitive domain [1,70]. However, in the literature, it has also been reported that acute physical exercises influence different cognitive domains differentially [1]. Our observation that the magnitude of the effect of acute physical exercises varies among different cognitive domains might be driven by specific mediators (e.g., study design-related factors such as the time of cognitive testing after exercise cessation) influencing the (positive) relationship between acute physical exercises and cognitive performance [2]. In this context, it can be hypothesized that SSREHIT might trigger neurobiological processes that lead to an improvement of attentional performance but that do not directly benefit the working memory performance. Given that the neurobiological mechanisms causing the exercise-induced improvement of cognitive performance are yet not fully understood [2], this explanation remains speculative and needs to be empirically proven (or rebutted) by future studies. However, even though the exact neurobiological mechanisms driving the exercise-related improvements of cognitive performance are yet not fully understood, it is undoubted that the cognitive enhancement relies on changes on different levels of analysis [2,71,72]. For instance, there is evidence in the literature showing that changes on a functional brain level (e.g., changes in cognition-related brain activity patterns) [42,73] or even on a molecular and cellular level (e.g., changes in the blood concentration of peripheral lactate [33,60] or brain-derived neurotrophic factor (BDNF) [74]) are associated with acute exercise-induced improvements in cognitive performance.

In line with previous studies in healthy younger adults [33,60] and our hypothesis, our repeated-measures correlation analysis indicated that changes in the peripheral blood lactate levels were linked to cognitive performance improvements in the exercise conditions. Thus, our findings buttress the idea that peripherally muscle-expressed lactate, which is able to cross the blood–brain barrier via monocarboxylate transporters, is utilized as “fuel” for cognitive processes [33,60,75,76,77,78,79,80,81,82,83,84]. In this context, it is worth noting that, in the study of Tsukamoto et al. [33], significant correlations between exercise-induced changes in the peripheral blood lactate concentration and executive functioning was noticed 20 min after exercise cessation (i.e., high-intensity interval exercise (HIIE) using four bouts of 4-min intervals at 90% VO_2peak_ that alternated with 3 min of active recovery at 60% VO_2peak_; total exercise duration of 33 min) but not immediately after or 10 min after exercise cessation [33]. In contrast to the study of Tsukamoto et al. [33], we observed significant correlations between exercise-induced changes in the peripheral blood lactate concentration and attentional performance 10 min after exercise cessation. This slightly divergent finding is perhaps related to differences in the exercise characteristics (SSREHIT vs. HIIE) and cognitive domains that were assessed (i.e., attentional performance vs. executive functioning). However, the finding of a positive relationship between exercise-induced changes in the levels of peripheral blood lactate concentrations and cognitive performance is not universal, given the observations (i) that, in healthy younger athletes (i.e., sprinter) five minutes after the cessation of a maximal multistage discontinuous incremental cycling test, higher levels of peripheral blood lactate concentrations and decreased attentional performance were noticed [85] and (ii) that, in healthy younger and older adults 15 min after the cessation of a multistage discontinuous incremental cycling test, negative correlations between the levels of peripheral blood lactate concentrations and executive functioning were reported [86]. The divergent findings between our study and the studies of Coco and colleagues [85,86] could be at least partly explained by (i) the differences in study methodology (e.g., exercise regime (SSREHIT protocol vs. maximal multistage discontinuous incremental cycling test), (ii) used cognitive test (d2 test vs. Attention and Concentration Task), time after exercise cessation (10 min vs. 5 or 15 min), and (iii) statistical analysis (repeated measures correlation vs. Pearson correlation). Given the mixed evidence in the literature, unarguably more critical examinations of the complex relationships between exercise prescription (e.g., exercise intensity), changes on different levels of analysis (e.g., changes on molecular and cellular level such as blood lactate levels), and cognitive performance are needed to broaden our understanding of the exercise–cognition interaction [72].

In addition, we observed that the participants provided higher ratings on the FS (indicating more pleasure) in the control condition than in the SSREHIT condition (i.e., after exercise and at posttest). However, our ratings in the SSREHIT condition are comparable to other studies performing SSREHIT (up to 8 × 5-s “all-out” effort sprints; ~37 s recovery between sprints) in healthy younger adults [19,22] and middle-aged adults with nondiabetic hyperglycemia [21]. In this context, it should be noted that the mean ratings on the FS immediately after exercise and posttest remained positive (1.40 and 1.55—correspond to “fairly good”), suggesting that SSREHIT does not induce strong displeasure. In addition, we did not observe time or condition effects regarding motivation, ability to concentrate, and mental fatigue, implying that these psychological variables did not have a strong influence on changes in our cognitive performance measures.

## 5. Limitations

The findings of this study should be interpreted by considering the following limitations. Due to the small sample size, we did not analyze whether sex influences the effects of acute SSREHIT on cognitive performance. There is some evidence that sex can influence the effects of acute physical exercises on cognitive performance [1], although this finding is not universal [87]. Given that this topic has not been exhaustively studied [88], future studies with a larger sample size are needed to investigate whether the effects of acute physical exercises (i.e., SSREHIT) on cognitive performance are moderated by biological sex. In this context, future studies should also consider comparing SSREHIT to control conditions other than seated rest (e.g., sham exercises or exercises with a different exercise intensity). Another inherent limitation of the SIT protocols (i.e., REHIT) is the lack of an appropriate and easy-to-assess physiological marker indicating that a participant performed sprints at an “all-out” effort. However, as done in previous studies, further research on acute SSREHIT should also consider to investigate changes in neurophysiological processes that might drive the acute improvements of cognitive performance in response to acute physical exercises such as changes on the molecular and cellular level (e.g., concentration changes in neurotrophic factors such as BDNF [74,89,90,91]) and/or on functional brain level (e.g., changes in electrophysiological signals assessed via electroencephalography [89,90,91,92,93] and/or cortical hemodynamics assessed via functional near-infrared spectroscopy or via functional magnetic resonance imaging; for reviews, see References [42,73]).

## 6. Conclusions

Our findings suggest that an acute bout of SSREHIT can improve specific aspects of attentional performance in younger adults without inducing strong displeasure. Thus, our study adds initial evidence to the literature that time-efficient exercise modalities such as SSREHIT can provide a sufficient stimulus to increase the performance in specific cognitive domains (e.g., attention). Furthermore, the current results also provide conforming evidence for a link between cognitive performance improvements and exercise-induced changes in peripheral blood lactate levels.

Further research that investigates the generalizability of our findings by studying the effect of SSREHIT on cognitive performance in other cohorts (e.g., older adults) and by considering further neurobiological mechanisms underlying cognitive performance improvements (e.g., changes in functional brain activity patterns) is necessary to substantiate our observations.

## Figures and Tables

**Figure 1 ijerph-19-00613-f001:**
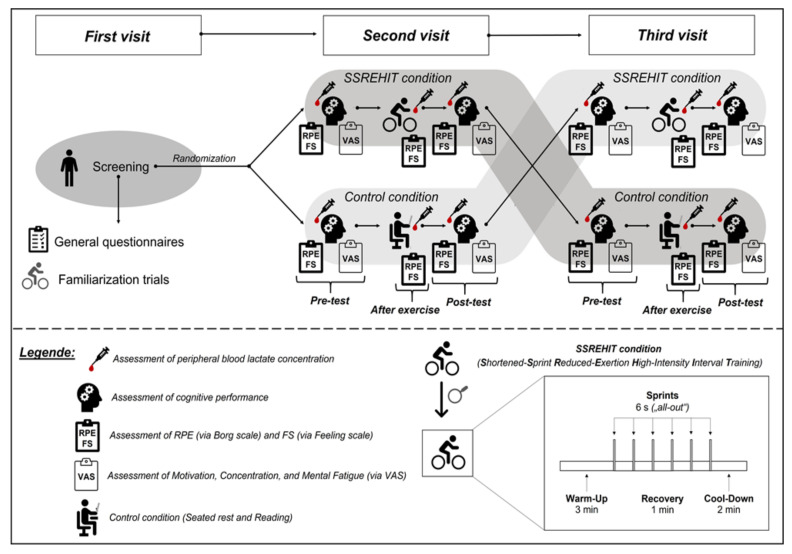
Schematic illustration of the study procedures. FS: Feeling scale, min: minute(s), s: seconds, SSREHIT: shortened-sprint reduced-exertion high-intensity interval training, RPE: Rating of Perceived Exertion (via Borg scale), and VAS: Visual Analogue Scale.

**Table 1 ijerph-19-00613-t001:** Overview of the general characteristics of the participants.

General Characteristics of the Participants	Mean ± Standard Deviation	Exercise Characteristics	Mean ± Standard Deviation
Education level (in years)	15.21 ± 1.69	(1) Sprint RPP (in W/kg)/PC (rpm)	12.90 ± 3.12/115.37 ± 14.37
BDI-II (total score)	3.63 ± 2.52	(2) Sprint RPP (in W/kg)/PC (rpm)	12.87 ± 2.91/115.11 ± 13.98
EHI (score)	89.21 ± 15.34	(3) Sprint RPP (in W/kg)/PC (rpm)	12.69 ± 2.63/114.37 ± 12.92
BSA PA (in min per week)	371.91 ± 218.18	(4) Sprint RPP (in W/kg)/PC (rpm)	12.34 ± 2.48/113.37 ± 12.41
BSA PE (in min per week)	292.30 ± 232.46	(5) Sprint RPP (in W/kg)/PC (rpm)	11.95 ± 2.34/112.37 ± 12.00
PSQI (global score)	5.11 ± 2.61 ^a^	(6) Sprint RPP (in W/kg)/PC (rpm)	11.88 ± 2.21/112.00 ± 11.14

BDI-II: Becks Depression Inventory-II (a score > 13 indicates depression [36]), BSA: Bewegungs-und Sportaktivitätsfragebogen (physical activity questionnaire), EHI: Edinburgh Handedness Inventory (a score of ≥50 indicates right-handedness [35]), kg: kilogram, min: minutes, PSQI: Pittsburgh Sleep Quality Index (a score of ≥6 indicates insomnia [41]), PA: physical activity, PE: physical exercise, PC: peak cadence, rpm: revolutions per minute, RPP: relative peak power (peak power divided by body mass), and W: watt. ^a^ Please note that only the data of 19 participants were analyzed, as one participant did not complete the full questionnaire.

**Table 2 ijerph-19-00613-t002:** Means (standard deviation) of the measures of cognitive performance.

	SSREHIT Condition			Control Condition		
	Pretest	Posttest	Delta Score	Pretest	Posttest	Delta Score
GZ (score)	558.90 (91.46)	608.63 (77.22)	49.74 (31.70)	563.58 (87.96)	596.16 (75.03)	32.58 (29.78)
SKL (score)	234.90 (48.27)	267.68 (41.55)	32.79 (21.92)	241.47 (48.62)	260.16 (39.52)	18.68 (16.60)
F% (score)	2.29 (2.34)	0.96 (1.05)	−1.33 (1.74) *	1.21 (1.16)	0.88 (0.99)	−0.33 (0.63) *
DSF (pts)	10.37 (1.64)	10.37 (1.89)	0.00 (1.89)	10.11 (1.66)	10.90 (1.80)	0.79 (1.36)
DSB (pts)	7.47 (2.09)	7.74 (2.02)	0.26 (1.52)	7.47 (2.17)	7.84 (2.12)	0.37 (1.46)

DSB: Digit Span Backward (higher score indicates better performance), DSF: Digit Span Forward (higher score indicates better performance), F%: number of all errors related to the total number of responses (lower score indicates better performance), GZ: total number of responses in the d2 test (higher score represents better performance), pts: points, SSREHIT: shortened-sprint reduced-exertion high-intensity interval training, and SKL: standardized number of correct responses minus errors of commission (higher score indicates better performance). *: Indicates a significant difference between conditions (*p* < 0.05).

**Table 3 ijerph-19-00613-t003:** Median (interquartile range) of the Rating of Perceived Exertion (RPE) scale, the Feeling scale, the Visual Analogue Scale (VAS) assessing motivation, concentration, and mental fatigue, and the peripheral blood lactate concentration levels.

	SSREHIT Condition			Control Condition		
	Pretest	After Exercise	Posttest	Pretest	After Exercise	Posttest
RPE (Borg scale)	6.00 (0.00)	16.00 (3.00) ^#^	11.00 (3.50) ^#,†^	6.0 (0.00)	6.00 (0.00) *	6.00 (0.00) *
Feeling scale	3.00 (2.50)	2.00 (1.00)	2.00 (3.00)	3.00 (2.00)	4.00 (2.00) *	3.00 (2.00) *
VAS (Motivation) (in mm)	80.00 (17.50)	n.a.	75.00 (17.00)	80.00 (15.00)	n.a.	80.00 (19.50)
VAS (Concentration) (in mm)	75.00 (27.50)	n.a.	75.00 (29.00)	70.00 (18.50)	n.a.	80.00 (15.00)
VAS (Mental fatigue) (in mm)	30.00 (37.50)	n.a.	30.00 (34.50)	30.00 (35.00)	n.a.	30.00 (30.50)
Peripheral blood lactate concentration (in mmol/L)	0.97 (0.49)	9.11 (6.03) ^#^	6.21 (6.19) ^#,†^	1.08 (0.61)	0.63 (0.50) *	0.57 (0.62) *^,#^

mm: millimeters; n.a.: not applicable, RPE: rating of perceived exertion, SSREHIT: shortened-sprint reduced-exertion high-intensity interval training, VAS: Visual Analogue Scale. Significant differences between both conditions are marked in the control condition. *: Indicates a significant difference between conditions at the same time point (*p* < 0.05). **^#^**: Indicates a significant effect of time compared to the pretest (*p* < 0.05). ^†^: Indicates a significant effect of time compared to after exercise (*p* < 0.05).

## Data Availability

The datasets generated during and/or analyzed during the current study are available from the corresponding author upon reasonable request.

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
