# Peer review of "The Influence of Acute Sprint Interval Training on Cognitive Performance of Healthy Younger Adults"

_ijerph, 2022, doi:10.3390/ijerph19010613_

Round 1
Reviewer 1 Report
The Influence of Acute Sprint Interval Training on Cognitive Performance of Healthy Younger Adults
This study addresses an interesting question by examining the effect of a time-efficient Sprint Interval Training, termed as shortened-sprint reduced-exertion high intensity interval training (SSREHIT), on cognitive performance. While the topic is of interest, I have some interrogations and reserves with some aspects of this study.
Introduction
I thought the introduction did not prepare the rest of the paper adequately. The authors did not convince me of the use of SIT training for their population. Although the authors present some studies stating the importance of enjoyment in the SIT protocols, these studies were not compared to a normal exercise protocol. Therefore, we can say that SSREHIT is enjoyable in within the SIT spectrum but expanding that conclusion to exercise adherence is not possible. Authors do not prepare the use of lactate in their project in the introduction or discuss any potential mechanisms between exercise and cognition.
Materials and methods
The study offers an interesting methodology but there are some major weaknesses.
My biggest concern was that authors did not measure resting maximal oxygen uptake (VO2max) or peak power output (PPO). Without this information, prescribing an exercise intensity seems very arbitrary. Knowing the participants baseline fitness scores can heavily influence their capacity to recover physiologically and perform well on a cognitive task. Without this information we can’t have an idea of the global fitness levels for your group.
Throughout their methodology the authors state that the participants did an “all out” sprint but how can, they be sure? How do we know participants gave supra max effort throughout the 6 sprints? Authors have no way of confirming this. Participants were not expected to maintain a wattage or a metabolic cost. What was done to motivate the participants? Were both sessions performed at the same time of the day?
Authors also did not take a baseline cognitive screening measure (i.e., MOCA, MMSE). Although it is a young group with higher education, we can’t rule out possibility of onset of vascular dementia or any other cognitive impairment.
Line 142: I believe this paragraph was supposed to be deleted.
Line 179: The authors do not justify the measure of blood LA but discuss it in the methodology. How do authors justify the timing of the LA test ? In response to “all-out” maximal exertion lasting 30-120 seconds, peak LA values of ≈15–25 mM may be observed 3–8 minutes postexercise.
There is lacking a regular exercise control group in this study. In order to fully support the results, authors should of included an exercise control group performing moderate intensity exercise and not compare SSREHIT to a static sedentary control group.
Statistical Analysis
Were all post-hoc t-tests Bonferroni-corrected for multiple comparisons?
Results
In my opinion, the results of this study add very little to the literature. Authors found very little changes following SSREHIT (cognitive or physiological). The result section is heavy and very difficult to follow.
Discussion
This is the part of the manuscript that I seem to have a little bit more difficulty with.
This discussion needs to be more focused.
The authors spend too much time discussing mechanisms in the discussion (compared to the introduction).
Overall this is a very innovative topic and one by which I am fully interested in. Unfortunately, this study had methodological flaws and very little to add to the literature in the state that it is presented. I wish you the best of luck moving forward.
Author Response
The Influence of Acute Sprint Interval Training on Cognitive Performance of Healthy Younger Adults
This study addresses an interesting question by examining the effect of a time-efficient Sprint Interval Training, termed as shortened-sprint reduced-exertion high intensity interval training (SSREHIT), on cognitive performance. While the topic is of interest, I have some interrogations and reserves with some aspects of this study.
Introduction
I thought the introduction did not prepare the rest of the paper adequately. The authors did not convince me of the use of SIT training for their population. Although the authors present some studies stating the importance of enjoyment in the SIT protocols, these studies were not compared to a normal exercise protocol. Therefore, we can say that SSREHIT is enjoyable in within the SIT spectrum but expanding that conclusion to exercise adherence is not possible. Authors do not prepare the use of lactate in their project in the introduction or discuss any potential mechanisms between exercise and cognition.
- We thank the reviewer for her/his feedback and carefully revised the introduction section taking the reviewers suggestions into account.. We revised our line of argumentation to better highlight the two main advantages of SSREHIT which can be summarized as follows: “Given the time-efficiency of SIT (as compared to continuous moderate-intensity exercise), in general, and the better psychological characteristics of SSREHIT (as compared to classical SIT using the Wingate protocol), in particular, SSREHIT can be considered as a promising exercise modality being worth future investigations [19,21,22].” (see lines 97 to 101 in the revised version of the manuscript) In this context, it should be kept in mind that younger adults perceived, among others, a lack of time as a significant barrier hindering them to perform physical exercises (Allison et al., 1999; Arzu et al., 2006; Gjestvang et al., 2020).
- Furthermore, we added the theoretical rationale why we have assessed changes in peripheral blood lactate concentration (see lines 113 to 128 in the revised version of the manuscript).
Materials and methods
The study offers an interesting methodology but there are some major weaknesses.
My biggest concern was that authors did not measure resting maximal oxygen uptake (VO2max) or peak power output (PPO). Without this information, prescribing an exercise intensity seems very arbitrary.
- We are thankful for the reviewer’s feedback. However, we are aware of the issues being related to different types of exercise intensity prescriptions (see our articles on exercise prescription (Gronwald et al., 2020; Herold et al., 2019; Herold, Törpel, et al., 2020)). In the current study, the exercise intensity prescription (see lines 199 to 219 in the revised manuscript) is based on existing studies in the field (S. B. Adamson et al., 2014; S. Adamson et al., 2014; S. Adamson et al., 2019; Haines, Broom, Gillibrand, & Stephenson, 2020; Haines, Broom, Stephenson, & Gillibrand, 2020). Notably, in all the above-mentioned studies the exercise intensity has been described as “all-out” efforts (S. B. Adamson et al., 2014; S. Adamson et al., 2014; S. Adamson et al., 2019; Haines, Broom, Gillibrand, & Stephenson, 2020; Haines, Broom, Stephenson, & Gillibrand, 2020). Moreover, in studies using, for instance, VO2 max. for a prescription of exercise intensity, the exercise intensity is typically controlled by measures of the heart rate which correspond to specific zones of VO2 max.. However, given the fact that measures of heart rate or other physiological measures typically used to characterize a specific exercise intensity (i.e., peripheral levels of blood lactate) are not suitable to prescribe the exercise intensity in SIT studies (e.g., measures of heart rate will not reach a reliable steady-state within a six-second sprint), it would be from little added value to perform a graded exercise test to establish VO2 max to set a specific exercise intensity in SIT (i.e. SSREHIT) protocols.
- Taking the above-mentioned evidence into account, we do not think that our prescription of exercise intensity constitutes a flaw but is rather bagged by the current literature (see lines 199 to 209 in the revised manuscript for detailed explanation).
Knowing the participants baseline fitness scores can heavily influence their capacity to recover physiologically and perform well on a cognitive task. Without this information we can’t have an idea of the global fitness levels for your group.
- We thank the reviewer for her/his thoughtful comment but only partly agree with her/his opinion. We agree with her/his view that the initial fitness level can influence the capacity to recover from exercise and might contribute to interindividual variability that can be observed concerning exercise-related changes of cognitive performance (see also our article on interindividual variability (Herold et al., 2021)). However, to the best of our knowledge, we are not aware of any acute exercise-cognition study that has addressed how the initial fitness level influences the recovery period (i.e., rest period after the cessation of the acute bout of physical exercise) before cognitive testing (i.e., post-exercise measurement). In other words, we are not aware of a study that provides a recommendation on the duration of the recovery period as a function of the initial fitness level (if the reviewer is aware of such a study it would be highly appreciated if she/he could share the reference with us). The duration of our recovery period (i.e., rest period after the cessation of the acute bout of physical exercise) of 10 minutes has been based on a meta-analysis showing that after such a time delay the greatest benefit on cognitive performance after vigorous-intensity physical exercise can be expected (Chang et al., 2012). This procedure has been described in lines 179 to 189 in the revised version of the manuscript.
- Although we did not provide data on the VO2 max., the PPO can be inferred from the first sprint (although we acknowledge that we do not directly label it in the manuscript as PPO) given the fact that PPO can be assessed using a six-second cycle test (Herbert et al., 2015). Furthermore, we provided data on the external load of every sprint (see Table 1) and on internal load (see Table 3). Providing data of external and internal load has been recommended in the literature to ensure transparency and allow for a comparison across studies (Gronwald et al., 2019; Herold, Törpel, et al., 2020). Moreover, we provided data on the regular physical activity level (see Table 1). Thus, we feel that we have provided sufficient data to characterize our participants and the exercise regime in our study in appropriate detail.
Throughout their methodology the authors state that the participants did an “all out” sprint but how can, they be sure? How do we know participants gave supra max effort throughout the 6 sprints? Authors have no way of confirming this. Participants were not expected to maintain a wattage or a metabolic cost. What was done to motivate the participants? Were both sessions performed at the same time of the day?
- We thank the reviewer for her/his thoughtful comment. We understand the concern of the reviewer that there is no guarantee that the participants did perform “all-out” efforts. Based on the facts (i) that almost all SIT studies with comparable exercise characteristics use this method to prescribe the exercise intensity (S. B. Adamson et al., 2014; S. Adamson et al., 2014; S. Adamson et al., 2019; Haines, Broom, Gillibrand, & Stephenson, 2020; Haines, Broom, Stephenson, & Gillibrand, 2020), (ii) that we have provided data concerning external load (see Table 1) and internal load (see Table 3) which is the recommended way to present exercise data (Gronwald et al., 2019; Herold, Törpel, et al., 2020), and (iii) that our data is comparable to other studies using a relatively similar protocol (Haines, Broom, Gillibrand, & Stephenson, 2020; Haines, Broom, Stephenson, & Gillibrand, 2020), we do not feel that a prescription of exercise intensity using “all-out” efforts is a flaw (see also our response to the previous comment). However, to acknowledge this limitation the following sentence has been added to the limitation section: “Another inherent limitation of SIT protocols (i.e., REHIT) is the lack of an appropriate and easy way to assess physiological markers indicating that a participant performed sprints with an “all-out” effort.”
- We added information about (i) how the participants were motivated: “Verbal encouragement was given during all sprints.” (see line 204 in the revised version of the manuscript) and (ii) what time of day the sessions took place: “The participants were asked to visit our laboratory three times at the same time of the day (±60 minutes)” (see lines 156 to 157 in the revised version of the manuscript).
Authors also did not take a baseline cognitive screening measure (i.e., MOCA, MMSE). Although it is a young group with higher education, we can’t rule out possibility of onset of vascular dementia or any other cognitive impairment.
- We thank the reviewer for this comment, but we kindly disagree with her/his opinion. The MMSE (Folstein et al., 1975) and MoCA (Nasreddine et al., 2005) are both instruments that have been validated as a screening tool for cognitive impairment in older adults but not in younger adults. Thus, the MMSE and MoCA are both not suitable to screen for cognitive impairment in younger adults although they both might have (mistakenly) been used for this purpose in other studies. Given the facts that (i) the prevalence of early-onset dementia in the age group of 30 to 34 years is 1.1 per 100 000 (Hendriks et al., 2021), (ii) the age range in our study is 20 to 28 years, and (iii) that the majority of participants were students, it is very unlikely that one of our participants suffered from early-onset dementia. Furthermore, a cognitive impairment would also be obvious in an abnormal performance in the D2 test or the digit span test. As we did not notice such abnormal performance, this reinforces our interpretation that our cohort of younger adults is cognitively healthy.
Line 142: I believe this paragraph was supposed to be deleted.
- Many thanks for your careful proofreading and for pointing out this mistake. We have now deleted these incorrect sentences.
Line 179: The authors do not justify the measure of blood LA but discuss it in the methodology. How do authors justify the timing of the LA test ? In response to “all-out” maximal exertion lasting 30-120 seconds, peak LA values of ≈15–25 mM may be observed 3–8 minutes postexercise.
- We thank the reviewer for her/his comment and revised this section as follows: “The selection for the timepoints of lactate assessment was based on previous and comparable studies in the field measuring peripheral blood lactate concentration before cognitive testing and after exercise cessation [33,60].” In this context, we wish to acknowledge that it is beyond the scope of this study to measure an exercise-related peak and/or a time course of lactate after exercise, rather we aimed at investigating the possible link between changes in lactate and changes in cognitive performance. Thus, we assessed peripheral levels of blood lactate in temporal proximity to the cognitive tests.
There is lacking a regular exercise control group in this study. In order to fully support the results, authors should of included an exercise control group performing moderate intensity exercise and not compare SSREHIT to a static sedentary control group.
- We thank the reviewer for her/his constructive feedback and added the following sentence in the limitation section (see lines 474 to 476 in the revised version of the manuscript): ”In this context, future studies should also consider to compare SSREHIT to control conditions other than seated rest (e.g., sham exercise or exercise with a different exercise intensity).” However, given the facts that (i) this study did not aim to investigate a possible dose-response relationship and (ii) that seated rest is an established control condition (Chang et al., 2012; Herold et al., 2018; Herold et al., 2021; Herold, Aye, et al., 2020; Pontifex et al., 2019), we do not feel that the lack of an active (exercise) control condition is a flaw.
Statistical Analysis
Were all post-hoc t-tests Bonferroni-corrected for multiple comparisons?
- We thank the reviewer for this comment. As stated in lines 295 to 296: “The adjustment of the alpha level of the post hoc test was conducted using the Holm correction (pholm).”
Results
In my opinion, the results of this study add very little to the literature. Authors found very little changes following SSREHIT (cognitive or physiological). The result section is heavy and very difficult to follow.
- We kindly disagree that we found very few changes. As described in our result section, we observed the following main findings:
- “The participants reduced the number of all errors (F% (t (18) = -2.247, p = 0.037, d = - 0.516 [CI 95% -0.989 to -0.030]) to a greater extent in the SSRHEIT condition as compared to the control condition.”
- “In addition, in the SSREHIT condition the following correlations were observed be-tween changes in peripheral blood lactate levels and measures of attentional performance: GZ (rm = 0.70 [CI 95% 0.342-0.878], p < 0.001), SKL (rm = 0.73 [CI 95% 0.391-0.891], p < 0.001), and F% (rm = -0.54 [CI 95% -0.802-0.093], p = 0.015). In the control condition, no significant correlations between changes in peripheral blood lactate levels and attentional performance were noticed (p > 0.05).”
- As we are, to the best of our knowledge, not aware of a comparable study that investigated the acute effects of SSREHIT on cognitive performance, we believe that these novel findings which are, in general, in accordance with the literature are worthy of being published and clearly add novel knowledge to the literature.
Discussion
This is the part of the manuscript that I seem to have a little bit more difficulty with.
This discussion needs to be more focused.
The authors spend too much time discussing mechanisms in the discussion (compared to the introduction).
- We believe that a discussion of potentially underlying mechanisms is essential to a scientific paper. We, therefore, are somewhat reluctant to substantially shorten the discussion. Noting the discussion might be too long for a “brief report” and following the suggestion of the other reviewer we now present our study as “research article”.
Overall this is a very innovative topic and one by which I am fully interested in. Unfortunately, this study had methodological flaws and very little to add to the literature in the state that it is presented. I wish you the best of luck moving forward.
References
Adamson, S., Kavaliauskas, M., Yamagishi, T., Phillips, S., Lorimer, R., & Babraj, J. (2019). Extremely short duration sprint interval training improves vascular health in older adults. Sport Sciences for Health, 15(1), 123–131. https://doi.org/10.1007/s11332-018-0498-2
Adamson, S., Lorimer, R., Cobley, J. N., Lloyd, R., & Babraj, J. (2014). High intensity training improves health and physical function in middle aged adults. Biology, 3(2), 333–344. https://doi.org/10.3390/biology3020333
Adamson, S. B., Lorimer, R., Cobley, J. N., & Babraj, J. A. (2014). Extremely short-duration high-intensity training substantially improves the physical function and self-reported health status of elderly adults. Journal of the American Geriatrics Society, 62(7), 1380–1381. https://doi.org/10.1111/jgs.12916
Allison, K. R., Dwyer, J. J., & Makin, S. (1999). Perceived barriers to physical activity among high school students. Preventive Medicine, 28(6), 608–615. https://doi.org/10.1006/pmed.1999.0489
Arzu, D., Tuzun, E. H., & Eker, L. (2006). Perceived barriers to physical activity in university students. Journal of Sports Science & Medicine, 5(4), 615–620.
Chang, Y. K., Labban, J. D., Gapin, J. I., & Etnier, J. L. (2012). The effects of acute exercise on cognitive performance: A meta-analysis. Brain Research, 1453, 87–101. https://doi.org/10.1016/j.brainres.2012.02.068
Folstein, M. F., Folstein, S. E., & McHugh, P. R. (1975). "Mini-mental state". A practical method for grading the cognitive state of patients for the clinician. Journal of Psychiatric Research, 12(3), 189–198. https://doi.org/10.1016/0022-3956(75)90026-6.
Gjestvang, C., Abrahamsen, F., Stensrud, T., & Haakstad, L. A. H. (2020). Motives and barriers to initiation and sustained exercise adherence in a fitness club setting-A one-year follow-up study. Scandinavian Journal of Medicine & Science in Sports, 30(9), 1796–1805. https://doi.org/10.1111/sms.13736
Gronwald, T., Bem Alves, A. C. de, Murillo-Rodríguez, E., Latini, A., Schuette, J., & Budde, H. (2019). Standardization of exercise intensity and consideration of a dose-response is essential. Commentary on "Exercise-linked FNDC5/irisin rescues synaptic plasticity and memory defects in Alzheimer's models", by Lourenco et al., published 2019 in Nature Medicine. J Sport Health Sci, 8(4), 353–354. https://doi.org/10.1016/j.jshs.2019.03.006
Gronwald, T., Törpel, A., Herold, F., & Budde, H. (2020). Perspective of Dose and Response for Individualized Physical Exercise and Training Prescription. Journal of Functional Morphology and Kinesiology, 5(3), 48. https://doi.org/10.3390/jfmk5030048
Haines, M., Broom, D., Gillibrand, W., & Stephenson, J. (2020). Effects of three low-volume, high-intensity exercise conditions on affective valence. Journal of Sports Sciences, 38(2), 121–129. https://doi.org/10.1080/02640414.2019.1684779
Haines, M., Broom, D., Stephenson, J., & Gillibrand, W. (2020). Influence of Sprint Duration during Minimal Volume Exercise on Aerobic Capacity and Affect. International Journal of Sports Medicine. Advance online publication. https://doi.org/10.1055/a-1255-3161
Hendriks, S., Peetoom, K., Bakker, C., van der Flier, W. M., Papma, J. M., Koopmans, R., Verhey, F. R. J., Vugt, M. de, Köhler, S., Withall, A., Parlevliet, J. L., Uysal-Bozkir, Ö., Gibson, R. C., Neita, S. M., Nielsen, T. R., Salem, L. C., Nyberg, J., Lopes, M. A., Dominguez, J. C., . . . Ruano, L. (2021). Global Prevalence of Young-Onset Dementia: A Systematic Review and Meta-analysis. JAMA Neurology, 78(9), 1080–1090. https://doi.org/10.1001/jamaneurol.2021.2161
Herbert, P., Sculthorpe, N., Baker, J. S., & Grace, F. M. (2015). Validation of a six second cycle test for the determination of peak power output. Research in Sports Medicine (Print), 23(2), 115–125. https://doi.org/10.1080/15438627.2015.1005294
Herold, F., Aye, N., Lehmann, N., Taubert, M., & Müller, N. G. (2020). The Contribution of Functional Magnetic Resonance Imaging to the Understanding of the Effects of Acute Physical Exercise on Cognition. Brain Sciences, 10(3), 175. https://doi.org/10.3390/brainsci10030175
Herold, F., Müller, P., Gronwald, T., & Müller, N. G. (2019). Dose-Response Matters! - A Perspective on the Exercise Prescription in Exercise-Cognition Research. Frontiers in Psychology, 10, 2338. https://doi.org/10.3389/fpsyg.2019.02338
Herold, F., Törpel, A., Hamacher, D., Budde, H., & Gronwald, T. (2020). A Discussion on Different Approaches for Prescribing Physical Interventions - Four Roads Lead to Rome, but Which One Should We Choose? Journal of Personalized Medicine, 10(3), 55. https://doi.org/10.3390/jpm10030055
Herold, F., Törpel, A., Hamacher, D., Budde, H., Zou, L., Strobach, T., Müller, N. G., & Gronwald, T. (2021). Causes and Consequences of Interindividual Response Variability: A Call to Apply a More Rigorous Research Design in Acute Exercise-Cognition Studies. Frontiers in Physiology, 12, Article 682891. https://doi.org/10.3389/fphys.2021.682891
Herold, F., Wiegel, P., Scholkmann, F., & Müller, N. G. (2018). Applications of Functional Near-Infrared Spectroscopy (fNIRS) Neuroimaging in Exercise⁻Cognition Science: A Systematic, Methodology-Focused Review. Journal of Clinical Medicine, 7(12), 1–43. https://doi.org/10.3390/jcm7120466
Nasreddine, Z. S., Phillips, N. A., Bédirian, V., Charbonneau, S., Whitehead, V., Collin, I., Cummings, J. L., & Chertkow, H. (2005). The Montreal Cognitive Assessment, MoCA: A brief screening tool for mild cognitive impairment. Journal of the American Geriatrics Society, 53(4), 695–699. https://doi.org/10.1111/j.1532-5415.2005.53221.x
Pontifex, M. B., McGowan, A. L., Chandler, M. C., Gwizdala, K. L., Parks, A. C., Fenn, K., & Kamijo, K. (2019). A primer on investigating the after effects of acute bouts of physical activity on cognition. Psychol Sport Exerc, 40, 1–22. https://doi.org/10.1016/j.psychsport.2018.08.015
Reviewer 2 Report
The presented results are important and interesting.
The authors take as a reference Wingate's SIT protocol of four bouts of 30 seconds "all-out" sprints intermitted by 4 minutes of recovery, which belongs to the classics of SIT, while there are several new proposals of protocols with all-out bouts shorter than 30 seconds, such as the work of Townsend, L.K. et al. Modified sprint interval training protocols. Part II. Psychological responses. Appl Phys Nutr Metab 2017, 42(4), 347-353. It is certainly worth enriching the Introduction with some detailed reports of the use of short SIT protocols important for variables such as engagement/enjoyment or perceived effort.
In the sentence about the purpose of the study, the authors should mention what characterizes their proposed SSREHIT in terms of bouts of effort and recovery bouts.
The Materials and Methods section is improperly structured; only used methods and statistical analyses were extracted while the whole section should be divided into subsections: 1/ participants - together with the eligibility and exclusion criteria; the characteristics of the participants in terms of education should also be included here (and not in the Results section), 2/ study design, here: highlighted in subsection Protocol description, 3/ physiological measures, 4/ psychological measures (here: cognitive as primary measures and others as secondary measures, e.g. here Pittsburgh index [for what purpose was it used? - this should be written], VAS, Borg), 5/ statistical analyses.
How numerous were the control and intervention trials in this study? Additionally, questions arise: How exactly was randomization performed? Did it take gender into account?
Were cognitive measurements made with parallel versions? These should be described. Parallel versions reduce the effect of recall, which is particularly important in measurements after short time intervals. If parallel versions were not used this should be added to Limitations, while Discussion seems too bold. It should be generalized or take into account factors such as the size of the intervention trials, gender of the subjects, etc. when citing previous studies.
The Limitations section should be separated from the Discussion.
There is some confusion about the classification of the manuscript as a Brief Report, because if the manuscript was enriched with at least the above-mentioned elements, the Research Methodology section was structured, and the discussion was conducted with thought and care, the manuscript could be classified as an Original Research Article. As a Brief Report, the manuscript has too general an Introduction, too detailed a description of the methodology, and too bold a discussion if parallel versions of the cognitive measures were not used. As a Brief Report the manuscript has far too many literature items. A Brief report requires particular precision of scientific language and synthetic presentation of content. Authors should decide whether to present a Brief report or Original Research and adjust the content to the appropriate framework.
The paper is characterized by extremely sloppy bibliographic notation, in many places there is no information about the pages of articles cited by the authors, and the notation of journal titles is not uniform either.
In Material and methods section there is a paragraph: The Materials and Methods should be described with sufficient details to allow others to replicate and build on the published results. Please note that the publication of your manuscript implicates that you must make all materials, data, computer code, and protocols associated with the publication available to readers. Please disclose at the submission stage any restrictions on the availability of materials or information. New methods and protocols should be described in detail while well-established methods can be briefly described and appropriately cited. This should be removed.
Best regards
Author Response
The presented results are important and interesting.
The authors take as a reference Wingate's SIT protocol of four bouts of 30 seconds "all-out" sprints intermitted by 4 minutes of recovery, which belongs to the classics of SIT, while there are several new proposals of protocols with all-out bouts shorter than 30 seconds, such as the work of Townsend, L.K. et al. Modified sprint interval training protocols. Part II. Psychological responses. Appl Phys Nutr Metab 2017, 42(4), 347-353. It is certainly worth enriching the Introduction with some detailed reports of the use of short SIT protocols important for variables such as engagement/enjoyment or perceived effort.
In the sentence about the purpose of the study, the authors should mention what characterizes their proposed SSREHIT in terms of bouts of effort and recovery bouts.
- We thank the reviewer for this comment and enriched the introduction by incorporating the suggested reference.
The Materials and Methods section is improperly structured; only used methods and statistical analyses were extracted while the whole section should be divided into subsections: 1/ participants - together with the eligibility and exclusion criteria; the characteristics of the participants in terms of education should also be included here (and not in the Results section), 2/ study design, here: highlighted in subsection Protocol description, 3/ physiological measures, 4/ psychological measures (here: cognitive as primary measures and others as secondary measures, e.g. here Pittsburgh index [for what purpose was it used? - this should be written], VAS, Borg), 5/ statistical analyses.
- We thank the reviewer for her/his helpful comment. According to the reviewer's comment, we restructured the Material and Methods section in accordance with her/his suggestions. Furthermore, we added the following sentence to clarify the purpose of assessing sleep quality via PSQI (see lines 170 to 175 in the revised version of the manuscript): “Given evidence (i) that in younger adults both regular levels of physical activity and measures of sleep are associated with cognitive performance (inhibitory control) and (ii) that sleep efficiency mediates the relationship between regular physical activity and cognitive performance (i.e., inhibitory control) [40], the BSA and the PSQI were assessed to provide a more comprehensive overview on the characteristics of the included participants (see Table 1)”
How numerous were the control and intervention trials in this study? Additionally, questions arise: How exactly was randomization performed? Did it take gender into account?
- We are thankful for the reviewer’s feedback. The experimental procedure is displayed in Figure 1 and every participant conducted the two experimental conditions once (SSREHIT and control). To clarify this, we have added the following sentence (see lines 188 to 189 in the revised version of the manuscript): “Each participant performed each condition (i.e., REHIT condition or control condition) once. ”
- Concerning randomization, the following is already stated in the manuscript (lines 186 to 187): “... we randomized the order of the SSREHIT and the control condition using a software (RITA version 1.51, Evidat, Germany).” As we did not aim to investigate sex-specific differences, we did not take gender into account in the randomization. In retrospect, the order of the REHIT and control condition was relatively balanced between the two sexes. Five (45%) of the 11 female participants started with the control condition and 6 (55%) with the REHIT session. Among the male participants, 5 (62%) started with the control condition and 3 (38%) with the REHIT session.
Were cognitive measurements made with parallel versions? These should be described. Parallel versions reduce the effect of recall, which is particularly important in measurements after short time intervals. If parallel versions were not used this should be added to Limitations, while Discussion seems too bold. It should be generalized or take into account factors such as the size of the intervention trials, gender of the subjects, etc. when citing previous studies.
- We thank the reviewer for this important hint and added the following (see also lines 257 to 259 in the revised manuscript): “For all cognitive tests, parallel versions were used to reduce the influence of learning (recall) effects.”
- With respect to the second part of the comment, we carefully revised the discussion section.
The Limitations section should be separated from the Discussion.
- We revised our manuscript according to the reviewer’s suggestion.
There is some confusion about the classification of the manuscript as a Brief Report, because if the manuscript was enriched with at least the above-mentioned elements, the Research Methodology section was structured, and the discussion was conducted with thought and care, the manuscript could be classified as an Original Research Article. As a Brief Report, the manuscript has too general an Introduction, too detailed a description of the methodology, and too bold a discussion if parallel versions of the cognitive measures were not used. As a Brief Report the manuscript has far too many literature items. A Brief report requires particular precision of scientific language and synthetic presentation of content. Authors should decide whether to present a Brief report or Original Research and adjust the content to the appropriate framework.
- We thank the reviewer for her/his valuable feedback and pointing out this mistake. The article type has been changed to “Research Article”.
The paper is characterized by extremely sloppy bibliographic notation, in many places there is no information about the pages of articles cited by the authors, and the notation of journal titles is not uniform either.
- We are thankful for the careful proofreading of the reviewer. We have corrected our bibliographic notations (e.g., missing information were added).
In Material and methods section there is a paragraph: The Materials and Methods should be described with sufficient details to allow others to replicate and build on the published results. Please note that the publication of your manuscript implicates that you must make all materials, data, computer code, and protocols associated with the publication available to readers. Please disclose at the submission stage any restrictions on the availability of materials or information. New methods and protocols should be described in detail while well-established methods can be briefly described and appropriately cited. This should be removed.
- We thank the reviewer for her/his careful proofreading and deleted those sentences.